# Hydrostructural Pedology, Culmination of the Systemic Approach of the Natural Environment

**Erik Braudeau [1,2,\*] and Rabi H. Mohtar [3,4]**

1   Institut de Recherche pour le Développement, (IRD), F13572 Marseille, France
2   Valorhiz—Parc Scientifique Agropolis, F34980 Montferrier sur Lez, France
3   Faculty of Agricultural and Food Sciences, American University of Beirut, Beirut 11-0236, Lebanon; mohtar@aub.edu.lb or mohtar@tamu.edu
4   Department Biological and Agricultural Engineering (BAEN), Zachry Department Civil and Environmental Engineering (CVEN), Texas A&M University, College Station, TX 77843, USA
\*   Correspondence: erik.braudeau@ird.fr or erik.braudeau@valorhiz.com; Tel.: +33-499-638-758

**Abstract:** The subject of this article is the dynamics of water in a soil pedostructure sample whose internal environment is subjected to a potential gradient created by the departure of water through surface evaporation. This work refers entirely to the results and conclusions of a fundamental theoretical study focused on the molecular thermodynamic equilibrium of the two aqueous phases of the soil pedostructure. The new concepts and descriptive variables of the hydro-thermodynamic equilibrium state of the soil medium, which have been established at the molecular level of the fluid phases of the pedostructure (water and air) in a previous article, are recalled here in the systemic paradigm of hydrostructural pedology. They allow access to the molecular description of water migration in the soil and go beyond the classical mono-scale description of soil water dynamics. We obtain a hydro-thermodynamic description of the soil's pedostructure at different hydro-functional scale levels including those relating to the water molecule and its atoms. The experimental results show a perfect agreement with the theory, at the same time validating the systemic approach that was the framework.

**Keywords:** pedostructure; systemic modelling; systemic variables; hydro-thermodynamic equilibrium; Gibbs free energy; fundamental thermodynamic variables; molecular; real and eulerian fluxes; hydric conductivity of the pedostructure

## 1. Introduction

The problem of water transfer equations in soil dates back to the beginning of soil science. The best-known equation and the basis of all models of water circulation in the soil, is the "Richards" equation, which associates Euler's law of continuity and Darcy's law extended to unsaturated soils:

$$\frac{d\theta}{dt} = -\frac{df}{dz} \text{ and } f = k\frac{dh}{dz} \tag{1}$$

where $\theta$ is the volumetric water content (unit less), $t$ is time, $f$ the flow, $z$ the vertical coordinate, $k$ the water conductivity and $h$ the soil water retention pressure.

We resume here the study of the water transfer equation in the soil with a completely new approach: that of the systemic approach we recently theorized [1–3] from the work of Bertalanffy, initiator of the general theory of systems [4] and Le Moigne [5], author of the General System model. The application of this systemic approach applied to pedology has created a new paradigm of characterization, water modeling, and representation of the natural environment (multi-scale mapping). It is named hydrostructural pedology [1–3] and is presented schematically in Figure 1.

Hydrostructural pedology allowed the development of a new physics of soil water, qualified as systemic, based on the recognition of the pedostructure and the two types of water associated with it [6–9]. We demonstrated that these two types of water in the pedostructure are two aqueous phases in pressure equilibrium ($h_{mi}(W_{mi}) = h_{ma}(W)$) and distinguished by their thermodynamic properties.

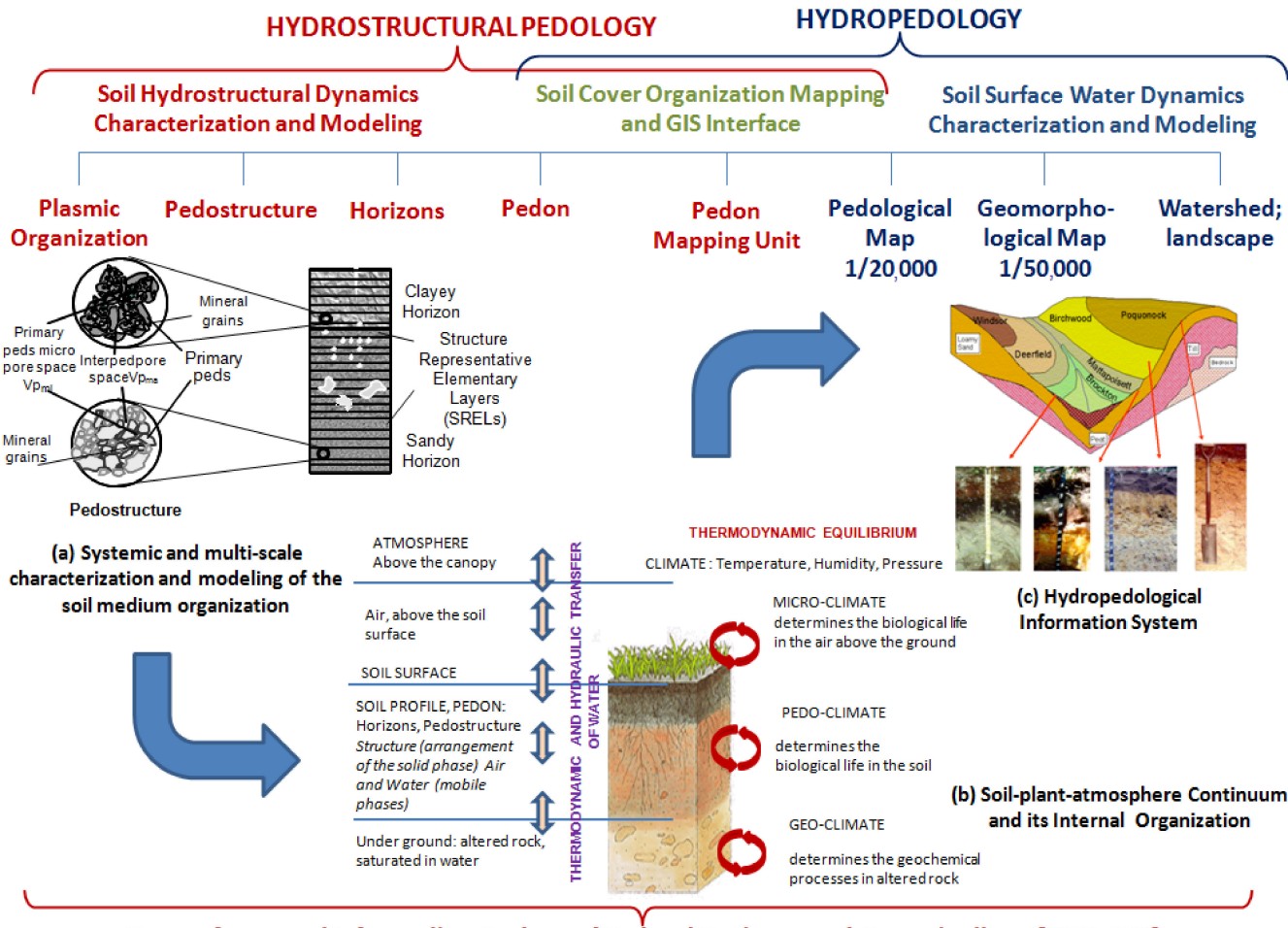

**Figure 1.** Place of hydrostructural pedology among the earth sciences [10].

This new physics of water in the soil has led to reconsideration the fundamental bases of classical thermodynamics; in particular, those of Gibbs free energy [11]. This study, associated with the new concept of "system organized into organized subsystems that are molecules and their atoms", has made it possible to develop a new vision of the thermodynamic equilibrium of the soil medium. The liquid and gas phases are all recognized as subsystems organized in molecules and themselves in atoms, relative to the solid phase that makes up the soil structure. Classical variables such as temperature, entropy, pressure, internal energy, and Gibbs free energy, can then be physically redefined and precisely explained accounting the two levels of organization: molecular and atomic.

In the present article, we will introduce these two levels of organization into the current description of the pedostructure organization for re-examining the terms of the Richards equation (Appendix A) after having rewritten it in the systemic framework of the hydrostructural pedology [1–3]. The new formulation will allow for taking into account all functional volumes as thermodynamics variables (volume of the whole system and volumes of hydro-functional subsystems). We will see then that the challenge becomes how to associate in the same equation an extensive variable (volume) and an intensive variable (potential). This is currently done empirically using Darcy's law extended to unsaturated

soils. In this article, we will see that the organization level, in which the extensive and intensive variables meet, is the molecular level of organization. At this level, the terms internal or external energy, internal or external pressure and internal or external chemical potential are all in equilibrium relationship, as explained in [11].

## 2. Updated Theoretical Background of Hydrostructural Pedology

### 2.1. The Pedostructure, Test Body of Hydrostructural Pedology

2.1.1. Preparation of a Standard Sample of Pedostructure

The pedostructure is the fundamental concept at the basis of hydrostructural pedology. Materially, it constitutes the first two levels of organization of the soil horizon: that of the clay plasma and that of the assembly of primary aggregates between them and possibly with other mineralogical or biological grains of sand size. Pedostructure is present in all soil horizons (Figure 2); its volume percentage in the horizon and its specific hydrostructural properties, due to the clay plasma that makes up the primary aggregates, characterize the hydrostructural behavior of a soil horizon.

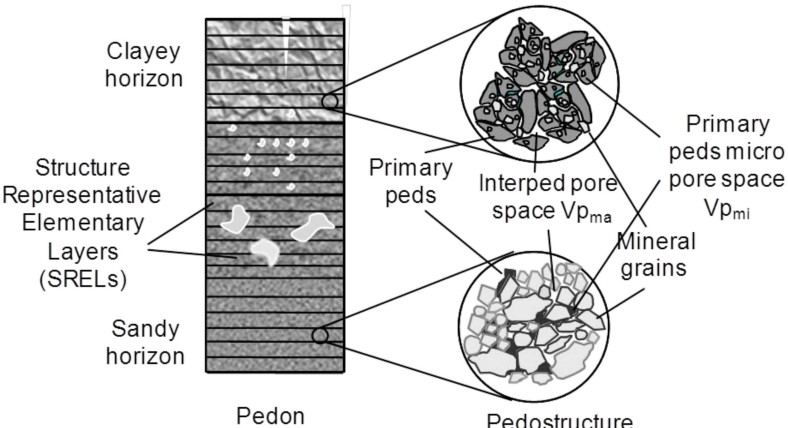

**Figure 2.** Representation of the internal organization of the pedon, hierarchized into its hydro-functional levels of organization: horizons, pedostructure, primary peds (after Braudeau et al. 2009 [12]).

The variables that are used for the systemic description of the pedostructure of any soil horizon are listed in Table 1. They all are reported to the same mass of solids: the pedostructural mass of the considered total soil volume.

**Table 1.** Pedostructure state variables. Subscripts *mi, ma* and *s*; refer to micro, macro and solids; *ip, st, bs* and *re*, refer to the name of the corresponding shrinkage phase of the shrinkage curve: interpedal, structural, basic and residual [12].

| Volume of Concern | Specific Volume [dm³/kg] | Specific Pore Volume [dm³/kg] | Specific Water Content [kg$_{water}$/kg$_{soil}$] | Non Saturating Water [kg$_{water}$/kg$_{soil}$] | Saturating Water [kg$_{water}$/kg$_{soil}$] | Suction [kPa] |
|---|---|---|---|---|---|---|
| Pedostructure | $\overline{V}$ | $\overline{Vp}$ | $W$ | | | $h$ |
| Interpedal porosity | | $\overline{Vp}_{ma}$ | $W_{ma}$ | $w_{st}$ | $w_{ip}$ | $h_{ma}$, $h_{ip}$ |
| Primary peds | $\overline{V}_{mi}$ | $\overline{Vp}_{mi}$ | $W_{mi}$ | $w_{re}$ | $w_{bs}$ | $h_{mi}$ |
| Primary particles | $\overline{V}_{s}$ | | | | | |

It is, therefore, necessary to define the representative sample of pedostructure in the laboratory: that sample upon which all the measurements of the hydro-functional curves of the soil will relate as well as the determination of their parameters. These characteristic curves are the shrinkage curve, $V(W)$, the water retention curve, $h(W)$, the unsaturated soil water conductivity, $k(W)$ and the swelling curve of primary aggregates as a function of time, $W_{mi}(t)$ [2,9].

In our study, a standard laboratory pedostructure sample is a soil sample that is reconstituted with what is traditionally called "fine earth", the 2 mm sieved soil from the fractionation of a moderately dry soil sample (<pF3) on a 2 mm sieve (can be 4 mm when the sample is very clayey with swelling clay). The fine earth is added layer by layer in a cylinder of 5 cm diameter and 5 cm height, placed on a damp terry cloth; each layer added wets along with the filling. The objective is to obtain a homogeneous sample in terms of structure and hydrostructural behavior. The soil cylinder is then subjected to 2 cycles of desiccation-humidification, the desiccation being carried out using either the Richards press at pF3 (15 bar) or evaporation in ambient air, the sample being positioned, in this case, so that the evaporation occurs simultaneously on both sides of the cylinder.

These preparation standards for the pedostructure sample are at the same level of importance as the oven temperature standard of 105° for the definition of dry soil. The term "pedostructural mass" is the mass of the solid phase that constitutes the pedostructure of the sample: it constitutes the universal benchmark for the extensive variables of a soil horizon (water content, salts etc. referred to the pedostructural mass).

### 2.1.2. Characterization and Modelling of the Hydrostructural Properties of the Soil

The parameters of the two equilibrium equations of the hydrostructural state of the pedostructure, the shrinkage curve $V(W)$ and the soil water retention curve $h(W)$, are determined from the curves measured on the standard sample using the TypoSoil® device which can simultaneously process up to 8 samples [12]. This characterization is totally accepted by the soil water model Kamel [9,13] and was fully established within the systemic paradigm of hydrostructural pedology with constant reference to the notions of nested organizations, hydro-functional levels of organizations (primary aggregate, pedostructure, soil horizon, pedon, primary soil unit, etc.), and using only variables, functions and parameters, said to be systemic because they are defined in the systemic description network of the hydrostructural pedology [1–3]. All the extensive variables of the studied homogeneous organized system, in particular the cut volume of the sample taken, are related to the fixed mass of the solid phase comprising the structure cut out in this volume.

However, the exact thermodynamic formulation of the water retention curve $h(W)$ at equilibrium of the two aqueous phases of the pedostructure [8] and, from this, the exact distribution of the two kinds of water content ($W_{ma}$ and $W_{mi}$) in the pedostructure as function of $W$, raises a new and important question about the descriptive variables of the model. This equation $h(W)$ links an extensive variable (water content) to an intensive variable (water suction). Indeed, we use a mini tensiometer (2 mm thick) planted laterally in a soil sample at depths z, to simultaneously obtain water suction ($h = h_{mi} = h_{ma}$), micro and macro water contents ($W_{mi}$ and $W_{ma}$) at thermodynamic equilibrium, locally in the same soil mini-layer. To answer the fundamental question of what exactly the spatial extension of is $W_{mi}$ and $W_{ma}$ corresponding to $h_{mi}$ and $h_{ma}$, we must recall the principal results of the previous article [11] on the hydro-thermodynamic equilibrium of the pedostructure at the molecular and atomic levels.

### 2.2. Molecular Thermodynamic Equilibrium of the Fluid Phases of the Pedostructure

2.2.1. Internal Molecular Organization of the Aqueous Phases at Equilibrium

The water molecule and simple gas molecules constitute the material point of the fluid phases of the natural environment. These molecules have a specific energy volume $V_m$, the sum of the volumes occupied by their constituent atoms, and a mass, the sum of the masses of their atoms. The free energy that the atom develops in the parent molecule is of the oscillatory type: $\frac{1}{2}m\lambda 2\nu 2$. The sum of the atomic free energies of the molecule constitutes the oscillatory energy of the molecule contained in its volume ($V_m = \sum V_{ai}$). It is this internal "free" energy of the molecule that has been identified as the temperature of a molecule. Thus, because of the mathematical property of fractions: $\left(\frac{a}{b} = \frac{c}{d} = \frac{a+c}{b+d}\right)$, not

only is the internal pressure equal everywhere in the molecule ($P = \frac{T}{V} = \frac{\sum T_{ai}}{\sum V_{ai}}$), but also the chemical potential defined by $\mu_m = \frac{T}{m} = \frac{T_{ai}}{m_{ai}} = \frac{\sum T_{ai}}{\sum m_{ai}}$.

In the aqueous or gaseous fluid phases in which the molecules, while being optionally ionized, maintain their chemical compositions in atoms, the internal oscillatory energies $\left(T = \frac{1}{2}m(\lambda v)^2\right)$ of the molecules are balanced with the kinetic energies acquired and maintained by what can be called thermal agitation ($E_{mv} = T(S-1) = \frac{1}{2}mv^2$): the "shocks" or meetings of molecules of the fluid phase between them. We can then associate each molecule with an occupancy volume: $V_{mt}$, that contains the two types of energy: oscillatory and linear kinetics (Figure 3).

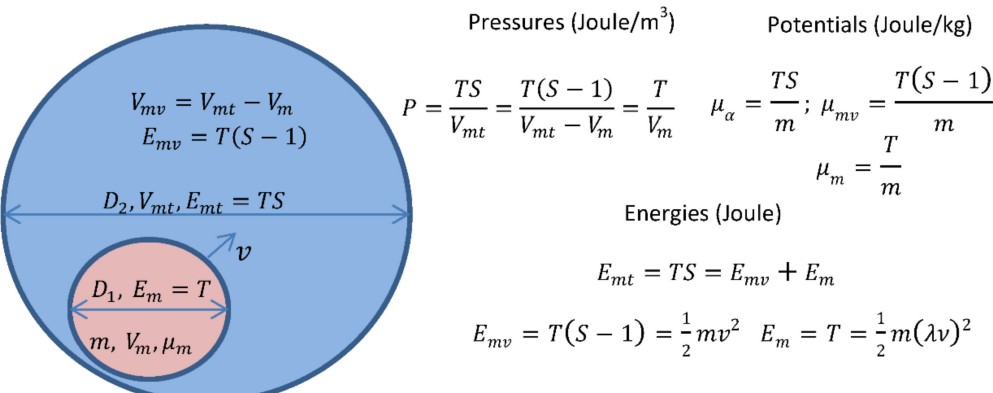

**Figure 3.** Conceptual model of the thermodynamic equilibrium at molecule scale; $v$ is the celerity of the molecule and $v_2$ is the number of shocks per unit of time [11]. The figure represents one molecule (in red) of volume $V_m$, in its occupational volume, $V_{mt}$, of diameter $D_2$ (red + blue). The external volume of the molecule, in blue, is $V_{mv} = V_{mt} - V_m$, the volume of its external energy: $E_{mv} = T(S-1) = \frac{1}{2}mv^2$. In a fluid phase, the volume concentration of energy, the pressure, is the same everywhere, in atoms, molecule and the intermolecular space.

At equilibrium, the volume concentrations of the internal and external energies (pressures) of the molecules are the same and this is where the molecular entropy of the phase ($S_a$) comes in. $S_a$ is a fractional number equal to $\frac{\sum V_{mt}}{\sum V_m}$, which makes thermodynamic equilibrium possible where all molecules have the same internal and external pressure.

Moreover, given that each molecule necessarily has the same chemical potential $\mu_{mi} = T_i/m_i$ as that of its atoms and the same mass volume, equilibrium is achieved if the chemical potential of the molecules is the same in all the phases. Molecules of chemical masses have different temperatures in the fluid phase in equilibrium, but have the same molecular chemical potential ($\mu_m = T_i/m_i$) and a chemical potential ($\mu_{m\alpha} = \mu_m S_\alpha$) that depends on the entropy ($S_\alpha$) of the $\alpha$ phase. We can define the chemical potential of the phase that concerns only the external kinetic energy of the molecules, which we will call the intermolecular chemical potential, $\mu_{v\alpha}$, (index $v$ void):

$$\mu_{v\alpha} = \mu_m(S_\alpha - 1) \tag{2}$$

The big difference with statistical thermodynamics is in the definition of temperature and the understanding of the free energy of the thermodynamic system (homogeneous liquid and gaseous fluid phases in equilibrium). The temperature of the phase is not the statistical average of a variable temperature around a value but, rather, the exact average of the temperatures of a finite number of molecules of different chemical species, the molecules of the same species having the same temperature (internal energy).

A phase is characterized by its entropy $S_\alpha$, an intensive state variable having the same value throughout the phase. At phase equilibrium, since the internal chemical potential of molecules, $\mu_m$, is the same everywhere, the overall molecular chemical potential of the phase, $\mu_\alpha = \mu_m S_\alpha$, is, therefore, an intensive variable characteristic of the phase. It is the

same for the external chemical potential of molecules of the phase ($\mu_{v\alpha}$), more specifically the characteristic kinetic potential of the phase ($\frac{1}{2}v^2$). This means that all molecules, regardless of their mass, have the same linear speed in the phase.

We, therefore, have a fundamental relationship between the chemical potential of the molecule, the entropy of the phase and the speed of the molecules in the phase:

$$\mu_{v\alpha} = \mu_m(S_\alpha - 1) = \mu_{m\alpha} - \mu_m = \frac{v_\alpha{}^2}{2} \tag{3}$$

where $v_\alpha$ is the linear speed of molecules of the phase at the thermodynamic equilibrium state. The pressure of the water molecules in this intermolecular space is $T_w(S_\alpha - 1)/(V_{mt_w} - V_{m_w})$, which is equal to the internal molecular pressure $T_w/V_{m_w}$ and the total molecular pressure in the phase: $(T_w S_\alpha)/V_{mt_w}$.

However, the suction pressure measured by the tensiometer in soil science, as shown previously [11], has the expression:

$$h = \rho_w(\mu_w - \mu_{w^\circ}) \tag{4}$$

where $\mu_{w^\circ}$ is the chemical potential of free water under air pressure and standard temperature. According to the Equation (3), and because $\mu_m$ is equal everywhere in all phases of the system at thermodynamic equilibrium, we can substitute the chemical potential of the water ($\mu_w$) by the intermolecular chemical potential ($\mu_{vw}$) without changing the value of the pressure $h$.

$$h = \rho_w(\mu_{vw} - \mu_{vw^\circ}) \tag{5}$$

This allows $\mu_{v\alpha}$ to be identified with the pressure potential of the water in the tensiometer, relating $h$ to the speed squared of molecules in the phase (Equations (3) and (5)).

The particularity of this speed is being the same for all molecules of the phase, whatever their mass, at thermodynamic equilibrium state since the criterion of intra and inter phase thermodynamic equilibrium is the molecular chemical potential ($\mu_m$) and not the temperature ($T_i$) as was shown in [11]. The temperature, identified with molecular energy, is in fact different for each chemical species of the phase since it is obtained from the molecular chemical potential $\mu_m$ of the fluid phases of the system in equilibrium ($T_i = m_i \mu_m$). The temperature of water molecules, for example, is equal to:

$$T_w = m_w \mu_m = m_w \frac{\mu_{m\alpha}}{S_\alpha} = m_w \frac{\mu_{v\alpha}}{S_\alpha - 1} \tag{6}$$

With these state variables of the thermodynamic equilibrium of the fluid phases (liquid and gaseous) of the pedostructure defined at the two organization levels, the molecule and the phase, we are able to search for the existing relationships between these newly defined variables and the usual ones (flow, water content, pressures, suction, etc.). It should be remembered that this link is only possible between variables defined according to the systemic approach, whether intensive or extensive.

We give in Table 2 the exhaustive list of hydro-thermodynamic variables qualified as systemic and which cover the four levels of organization: macroscopic and microscopic of the aqueous phase then molecular and atomic of the phase.

**Table 2.** Primary thermodynamic variables and their units.

| Variables | Symbols | Equation | Type | Units |
|---|---|---|---|---|
| Total molecular energy | $E_{mt}$ | $\frac{1}{2}\left[m(\lambda v)^2 + mv^2\right]$ | Energy | joule |
| Temperature | $T = E_m$ | $\frac{1}{2}m(\lambda v)^2$ | Energy | joule |
| Entropy S | $S$ | $V_{mv}/V_m$ | inter-mol. Vol./molecular Vol. | number |
| Pressure P | $P = P_m = P_{mv} = P_{mt}$ | $\frac{T}{V_m}$, $\frac{T(S-1)}{V_{mv}}$, $\frac{TS}{V_{mt}}$ | Energy/Volume | joule/m³; Pa |
| Chemical potential μ | $\mu_m$, $\mu_{mv}$, $\mu_\alpha$ | $\frac{T}{m}$, $\frac{T(S-1)}{m}$, $\frac{TS}{m}$ | Energy/Mass | joule/kg |

Temperature is pseudo-intensive, the others are true intensive variables and they represent each point of the medium and stay unchanged across scales [11].

2.2.2. Identification of Constants $\overline{E}_{ma}$ and $\overline{E}_{mi}$ as Intermolecular Free Energies of the Pedostructure

We know that at the macroscopic level of the phases of the pedostructure, the free energies of the two aqueous phases of the pedostructure ($\overline{E}_{mi}$ and $\overline{E}_{ma}$) are observed constant despite a change in the water content of this phase in the defined system of the pedostructure. Following Sposito [13], $\overline{E}_{mi}$ and $\overline{E}_{ma}$ was identified before as the free energy $\overline{G}_{mi} = \mu_{wmi}W_{mi}$ and $\overline{G}_{ma} = \mu_{wma}W_{ma}$, $W_{mi}$ and $W_{ma}$ being the water contents micro and macro of the pedostructure. Now, following our previous study [11], which differentiates the intermolecular energy ($T(S_\alpha - 1)$) from the total energy ($TS_\alpha$), $\overline{E}_{mi}$ and $\overline{E}_{ma}$ are defined as the intermolecular energy, corresponding to kinetic energy of molecules:

$$\overline{E}_{wma} = \frac{1}{2}\overline{m}_{wma}v_{ma}{}^2 = \mu_{vma}\overline{m}_{wma} = \overline{n}_{wma}m_w\mu_{vma} \tag{7}$$

$$\overline{E}_{wmi} = \frac{1}{2}\overline{m}_{wmi}v_{mi}{}^2 = \mu_{vmi}\overline{m}_{wmi} = \overline{n}_{wmi}m_w\mu_{vmi} \tag{8}$$

This relationship involves the number of water molecules $\overline{n}_{w\alpha}$ in the aqueous phase "$w\alpha$" (macro or micro), whose molecular mass is $m_w$. The horizontal line above the extensive variables signifies the ratio to the solid mass present in the same elementary volume.

Recall that these two aqueous phases coexist in the section (elementary layer) at $z$, one surrounding the other, in the pedostructure sample. They are indexed: $W_{ma}$ and $W_{mi}$. The fact that the energy of the phase $\alpha$ (macro or micro), $\overline{E}_{w\alpha}$, is constant despite a change in phase's water content ($\overline{m}_{w\alpha}$) or in its chemical potential ($\mu_{w\alpha}$), appears as a displacement constraint for the molecules since a potential gradient is created as soon as a deficit of water molecules appears in the system. This constraint is written:

$$d\overline{E}_{w\alpha} = 0 = \overline{m}_{w\alpha}d\mu_{v\alpha} + \mu_{v\alpha}d\overline{m}_{w\alpha} = 0 \tag{9}$$

or, for each $\alpha$ aqueous phase (macro or micro):

$$\frac{d\mu_{v\alpha}}{\mu_{v\alpha}} = -\frac{d\overline{m}_{w\alpha}}{\overline{m}_{w\alpha}} \text{ similar to } d\log(\mu_{v\alpha}) = -d\log(\overline{m}_{w\alpha}) \tag{10}$$

As the soil medium of the pedostructure is in thermodynamic equilibrium, we have, at every point of the medium, equality of the retention pressures between the two phases:

$$h_z = h_{ma} = h_{mi} = \rho_w(\mu_{vma} - \mu_{vma^\circ})_z = \rho_w(\mu_{vmi} - \mu_{vmi^\circ})_z \tag{11}$$

This gives the following general equation, since $\mu_{vma} - \mu_{vma^\circ} = \mu_{vmi} - \mu_{vmi^\circ}$ and according to (10):

$$\mu_{vma}d\log(\overline{m}_{wma}) = \mu_{vmi}d\log(\overline{m}_{wmi}) = -d\mu_{vma} = -d\mu_{vmi} \tag{12}$$

Note that this equation is valid only if the saturated state corresponding to $h = 0$ is set and therefore the potential $\mu_{vma^\circ}$ and $\mu_{vmi^\circ}$ corresponds to $W_{ma^\circ} = W_{ma_{sat}}$ and $W_{mi^\circ} = W_{mi_{sat}}$ and that all these equilibrium equations are deduced from the fact that the "free" energies $\overline{E}_{wma}$ and $\overline{E}_{wmi}$ of the two aqueous phases in equilibrium, defined and expressed by the fundamental expressions (7) and (8), are constant and characteristic of the pedostructure.

2.2.3. Definition of the Molecular Flux $f_{ma}$ in the Pedostructure in Thermodynamic Equilibrium

Consider the two aqueous phases of the pedostructure, one in the inter-aggregate space (macro phase), surrounding the other (micro phase) in the clay plasma of the primary peds. Only the macro phase is in contact with air and in capillary continuity throughout the sample. It has the possibility of moving according to a potential gradient created by the departure of water molecules from the surface. The micro phase has no capillary continuity

and is in contact only with the macro phase with which it locally comes into pressure equilibrium by exchanging water molecules.

In the absence of a potential gradient, the sum of the "speed" vectors of all molecules is zero (zero divergence). When a potential gradient is set up, on the z axis for example, an acceleration field automatically derives from the potential and is applied to the molecules placed in this gradient; in the case of the macro phase:

$$\gamma_{ma_z} = \frac{d\mu_{vma}}{dz} \tag{13}$$

In the case of our standard experiment of drying a soil cylinder by evaporation of water on the surface, the force exerted on all molecules of the phase is, in accordance with Newton's second law: $\vec{F}_i = m_i \vec{\gamma}_{wma}$. This force determines the elementary pressure carried by each molecule of the phase on its environment: ($p = \rho_w \gamma_{wma}$; cf. [11]), which causes a molecular flux $f_{ma}$ through the surface ($s_{ma}$) (occupied by the macro aqueous phase) of the sample section at depth $z$, such as:

$$\gamma_{ma_z} = \frac{df_{ma}}{dt} = \frac{d\mu_{vma}}{dz} \tag{14}$$

where $f_{ma}$ is the molecular flux of the macro phase, identified to the molecular speed of phase molecules along a gradient line if it exists (non-zero divergence). It is, therefore, an intensive phase variable that has the same value for each molecule in the phase. This is what conceptually distinguishes this type of (molecular) flux from the flows usually considered as flow in the Euler equation, defined as the speed of passage of a volume of n molecules through a chosen surface and not the actual passage of molecules. Moreover, as said above, the argument of velocity vectors of molecules of a phase is equal throughout the phase in thermodynamic equilibrium, whatever their molecular masses.

## 3. Materials and Methods

### 3.1. Soils

#### 3.1.1. Provenance

All tested soils in this study come from Martinique; they were collected and characterized as part of a IRD project to establish a SIG of Soils of the Martinique [14]. The goal of the project was to physically characterize the hydrostructural properties of the soils described in the notice of the very detailed existing soil map of Martinique. Soils are clayey of volcanic origin, differentiated by pedogenesis according to the geomorphological situation and geographical position they occupy around the ancient volcano. These lead, over small distances, to well-differentiated pedohydric regimes and different microclimates and plant cover on the surface.

#### 3.1.2. Hydrostructural Characterization

Hydrostructural characterization consists of measuring the shrinkage curve and the potential curve performed at the same time on the same sample [7]. At the time of the project, TypoSoil did not exist yet and the hydrostructural characterization was accomplished by measuring the shrinkage curve $V$ ($W$) and the suction curve $h$ ($W$) (or water retention) of the soil on two separate samples. The porous ceramic of the tensiometer had to be placed in the center of the sample, and the water had to evaporate uniformly over the entire surface so that the curve was representative of the sample.

### 3.2. Measuring Apparatus of the Hydric Conductivity of the Pedostructure

The apparatus used was manufactured to measure the water conductivity of the soil pedostructure (Figure 4). It is composed of a balance on which rests a metal cylinder of 5 cm in diameter and 5 cm in height containing the soil sample, collected at a moisture state close to the field capacity. The cylinder is provided with two holes that allow the

introduction of two mini-tensiometers T1 and T2 (diameter 2 mm), positioned 1 cm and 2 cm, respectively, from the surface. The sample, reshuffled or not, is first brought to saturation with its cylinder on a sandbox. The upper surface is then leveled at the edge of the cylinder while the lower surface is covered with plastic film to prevent any evaporation on this side. Finally a flat ring, with an outer diameter equal to the diameter of sample and forming a strip a few millimeters in width, is laid on its upper face to limit the lateral evaporation that occurs after reduction in the diameter by shrinking.

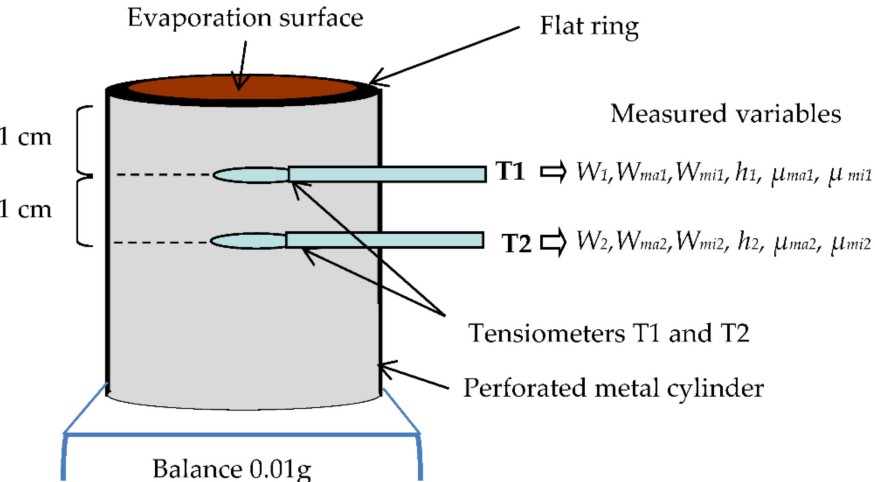

**Figure 4.** Device for measuring the hydric conductivity of the pedostructure.

The tensiometers are connected to a box of pressure sensors, in turn connected to a computer. The assembly is placed in a thermostatic chamber at 34 °C. Tensions and weights are recorded every 5 min. We deduce the values of the overall water content $(W_t)$, the water potential $(\mu_{mi}, \mu_{ma})$ of the micro and macro phases at two positions of the tensiometer as well as the corresponding local water contents $(W_{mi}, W_{ma})$ by using the following relationships:

$$W_t = \frac{M_t - M_S}{M_S}; \ h = h_{ma} = h_{mi} = \rho_w(\mu_{ma} - \mu_{ma\circ}) = \rho_w(\mu_{mi} - \mu_{mi\circ}) \tag{15}$$

$$h = \rho_w\overline{E}_{ma}\left(\frac{1}{W_{ma}} - \frac{1}{W_{maSat}}\right) = \rho_w\overline{E}_{mi}\left(\frac{1}{W_{mi}} - \frac{1}{W_{miSat}}\right) \tag{16}$$

$$\overline{E}_{ma} = W_{ma}\mu_{Wma} \text{ and } \overline{E}_{mi} = W_{mi}\mu_{Wmi} \tag{17}$$

where $\overline{E}_{mi}, \overline{E}_{ma}, W_{maSat}$ and $W_{miSat}$ are the characteristic parameters of the retention curve $h(W)$ given by the previous hydrostructural characterization of a soil horizon.

### 3.3. Systemic Variables Used for Modelling the Water Movement at the Different Organization Levels of the Pedostructure

#### 3.3.1. Hierarchical Arrangement of the Solid, Liquid and Air Phases at a Given Depth

The standard object of the study being a sample of pedostructure of cylindrical shape by which we study the variation of the water state along the $z$ axis, all the descriptive variables used must relatable to the same level of scale: that of the horizontal section of the sample at depth $z$, then allowing us to relate the two microscopic and macroscopic aspects of the sample. We can imagine this section, over an elementary height $dz$, surfaces occupied by the well identified phases: the solid phase $(s_s)$, two aqueous phases $(s_{mi})$ and $(s_{ma})$, and the gas phase $(s_{air})$ such that the total surface of the section $s_t$ is the sum $s_t = s_s + s_{mi} + s_{ma} + s_{air}$. The entire surface is homogeneously filled with these 4 phases with the essential constraint that the arrangement of the 4 phases between them is respected:

the solid phase is surrounded by micro water, which is surrounded by the macro phase, and the macro phase surrounded by the air phase.

Recall that the basic assumption of the systems approach is to consider the solid phase of the structure as invariant in mass in the discretization of space: the elementary horizontal layers of volume $\delta V = s_t dz$ all contain the same quantity of structural mass.

$$\frac{\delta m_s}{dz} = \frac{\rho_s s_s dz}{dz} = \rho_s s_s = \frac{M_s}{L} \tag{18}$$

where $\rho_s s_s$ is a fixed characteristic of the homogeneous sample in terms of structure and its structural mass $M_s$; $\rho_s$ is the actual density of the solid phase, $M_s$ is the total mass of the dry sample and $L$ is its length.

The extensive variables, such as water contents and organized volumes and sub-volumes ($W$, $W_{mi}$, $W_{ma}$, $\overline{V}$, etc.), are all related to the mass of the local structural phase: $\delta m_s = \rho_s s_s dz$, the mass contained in the same volume $\delta V$ as that in which the other variables are defined. We have for example:

$$W_z = \frac{\rho_w s_w dz}{\rho_s s_s dz} = \frac{\rho_w s_{wz}}{\rho_s s_{sz}} \tag{19}$$

In the systemic approach, this structural mass is the fixed reference to which are attached all the variables of the same volume and which, in turn, is variable with the water content. This ensures consistency in the creation and definition of descriptive variables as well as their correct use.

### 3.3.2. Extensive Variables such as Volumes and Water Contents

The crucial problem is indeed that defining the extensive variables of water content and of volumes that depend on the organizational scale at which the variable is considered: the macroscopic level of the entire system ($V_t$, $W_t$, etc.) or a discretized subsystem between two depths ($V_{z1-z2} = \Delta V_{1-2}$) or that considered at the molecular level, that of the horizontal section at depth $z$.

(1) The global variables $W_t$, $V_t$, etc. marked with the index $t$, are considered homogeneous and in thermodynamic equilibrium over the entire pedostructural system, the subject of the study. The volume ($V_t$) divided by the total mass of the solid phase ($M_{st}$) constituting the structure of the system, is the mass volume, which we write as $\overline{V}_t$: $\overline{V}_t = V_t / M_{st}$. Likewise, the overall water content of the system, $W_t$, is the mass water content of the system ($M_w$) divided by the structural mass ($M_s$) and written as: $W_t = M_w / M_{st}$. These variables, all related to the same structural mass, are additive: $W_t = W_{ma_t} + W_{mi_t}$. We have the following equalities:

$$W_t = \rho_w \frac{V_{wt}}{M_{st}} = \rho_w \overline{V}_{wt} = \frac{M_w}{M_{st}} \tag{20}$$

The mass volume of the pedostructure (mass pedostructural volume) is variable with its water content in accordance with a characteristic property of the soil: its shrinkage curve: $\overline{V}_t = f(W_t)$.

(2) A second type of variable is the local variable, defined for a delimited part of the homogeneous medium of the total system. In our case of a standard sample of pedostructure, this is a horizontal layer of the sample between the dimensions z1 and z2. This layer is a subsystem of the overall system, and has the same structural characteristics but the extensive variables of volume and water content of the various mobile phases of the system (aqueous and gaseous) are not related to the total structural mass of the system. They are related to the local structural mass of the medium, between the depths z1 and z2. This type of variable with an extensive character is indexed with $\Delta z$

$$W_{\Delta z} = \rho_w \frac{V_{w\Delta z}}{M_{s\Delta z}} = \rho_w \overline{V}_{w\Delta z} = \frac{M_{w\Delta z}}{M_{s\Delta z}} \tag{21}$$

(3) The third category of variables of an extensive nature is the "molecular" variable, a function of $z$. These variables are attached to the horizontal section of the sample identified by the corresponding z score. Let us redefine the mass volumes, like $\overline{V}_z$, using all the descriptive variables of the aqueous and gas phases defined at this depth z:

$$\overline{V}_z = \lim_{dz \to 0} \frac{s_t}{\rho_s \, s_s} \frac{dz}{dz} = \frac{s_t}{C} \equiv M_s^{-1} L^3 \tag{22}$$

where

$$C = \frac{\rho_s \, s_s \, dz}{dz} = \frac{dm_s}{dz} = \rho_s \, s_s \text{ at depth z.} \tag{23}$$

Subsequently, the $z$ index, indicating that the variable is molecular, will generally be omitted.

Assuming that the medium is homogeneous from the perspective of its structure, we can consider c as a constant that can be estimated at $C = \frac{dm_s}{dz} = \frac{M_s}{L}$ for a cylindrical sample of height L and structural mass $M_s$. We define the water content variables in the same way:

$$W_{ma} = \frac{\rho_w}{\rho_s} \frac{s_{ma}}{s_s} = \frac{\rho_w}{C} s_{ma} = \rho_w \overline{V}_{ma} \tag{24}$$

$$W_{mi} = \frac{\rho_w}{\rho_s} \frac{s_{mi}}{s_s} \frac{dz}{dz} = \frac{\rho_w}{C} s_{mi} = \rho_w \overline{V}_{mi} \tag{25}$$

$$W = \frac{\rho_w}{\rho_s} \frac{s_w}{s_s} \frac{dz}{dz} = \frac{\rho_w}{C} s_w = \rho_w \overline{V}_w \tag{26}$$

Note that, since $\rho_w$ and $C$ are constants, any ratio of two occupied surfaces of a section in z ($s_{ma}/s_{mi}$ at depth z, for example) is equal to the ratio of the volumes based on these surfaces and for height dz. This ratio of two differentiated surfaces at the molecular level of a horizontal section of the sample, at z, can be considered equivalent to the ratio of the corresponding extensive variables at the same z, for example:

$$\frac{s_w}{s_t} = \frac{V_w}{V} = \frac{\overline{V}_w}{\overline{V}} = \theta_w = \frac{W}{\rho_w \overline{V}} \tag{27}$$

where $s_w$ is the area of the section at z occupied by water ($s_w = s_{ma} + s_{mi}$) and $s_t$ is the total area, at $z$.

3.3.3. Concomitant Variation of Organizational and Fluxes Variables at $z$

The two water fluxes, $fe_w$ and $f_w$, are, according to their physical definition (30) and (31), proportional to the molecular flux according to:

$$\rho_w \overline{V} fe_w = W_{ma} f_w = W f_{ma} \tag{28}$$

or, dividing by $\rho_w \overline{V}$:

$$fe_w = \theta_{wma} f_w = \theta_w f_{ma} \tag{29}$$

However, these Equations (32) and (33) do not provide any information on their reciprocal variations in time and space: it must be the same $dz$ for the same $dt$ that makes up the equations defining the three types of variable containing the equation (flux, volume and mass). It is possible to resolve the uncertainty about $dz$ by considering the derivative of these variables with respect to z. The derivative of $f_{ma}$ defined by Equation (31) gives:

$$\frac{df_{ma}}{dz} = \frac{d}{dz}\left(\frac{s_{ma}}{s_{ma}} \frac{-dz}{dt}\right) = -\frac{d}{dt}\left(\frac{s_{ma} dz}{s_{ma} dz}\right) = -\frac{dW_{ma}}{dt W_{ma}} = -\frac{d \ln W_{ma}}{dt} \tag{30}$$

The change in sign results from the fact that one passes from a variation of speed, $\frac{dz}{dt}$, to a variation of volume calculated with the height $dz$ taken in the opposite direction of the speed. In the same way, we also have:

$$\frac{df_w}{dz} = \frac{d}{dz}\frac{s_w}{s_{ma}}\frac{dz}{dt} = \frac{s_w}{s_{ma}}\frac{d}{dz}\frac{s_w}{s_w}\frac{dz}{dt} = -\frac{W}{W_{ma}}\frac{d\ln W}{dt} = -\frac{dW}{W_{ma}dt} \tag{31}$$

$$\frac{df e_w}{dz} = \frac{d(s_w dz)}{dz\, s_t\, dt} = -\frac{d\ln W}{dt}\frac{s_w}{s_t} = -\frac{d\ln W}{dt}\frac{W}{\rho_w \overline{V}} = -\frac{dW}{\rho_w \overline{V} dt} \tag{32}$$

Note that the relations (34) to (36) participate in the definition of local extensive variables (at depth z); we have in fact:

The indeterminacy having been lifted, it is allowed to relate these 3 equations to each other, which gives:

$$\frac{df_w}{dz} \bigg/ \frac{df_{ma}}{dz} = \frac{dW}{dt} \bigg/ \frac{dW_{ma}}{dt} = \left(1 + \frac{dW_{mi}/dt}{dW_{ma}/dt}\right) \tag{33}$$

$$\frac{df e_w}{dz} \bigg/ \frac{df_w}{dz} = \frac{W_{ma}}{\rho_w \overline{V}} = \theta_{wma} \tag{34}$$

and we can rewrite the continuity equation in this form:

$$\frac{dW}{dt} = -\rho_w \overline{V}\frac{df e_w}{dz} = -W_{ma}\frac{df_w}{dz} = -W_{ma}\frac{df_{ma}}{dz}\left(1 + \frac{dW_{mi}/dt}{dW_{ma}/dt}\right) \tag{35}$$

Other relationships between fluxes and water contents are given in Appendix B.

### 3.3.4. Spatial Variation of the Product $f_{ma}W_{ma}$

We saw that $f_{ma}$ is a molecular flux of the aqueous phase macro, and $W_{ma}$ is the water content of this phase at depth z, given by Equation (24). The problem is the constant C that makes reference to the solid phase. By taking the correct expression for the solid phase, we can then consider $f_{ma}$ as the speed of each molecule and $f_{ma}W_{ma}$ as the concentration of momentum whose derivative with respect to time is a force.

Consider the molecular expression of the product: $f_{ma}W_{ma}$, in accordance with Equations (24)–(27) and (31) of the physical definition of the two variables and their derivatives with respect to z:

$$f_{ma}W_{ma} = \frac{dz}{dt}\frac{\rho_w s_{ma}}{C} = \frac{\rho_w s_{ma}dz}{\rho_s\, s_s\frac{dz}{dz}dt} = -\frac{\rho_w \delta V_{ma}}{\frac{\delta m_s}{dz}dt} = -\frac{\rho_w dV_{ma}}{\alpha_z m_{s_{zma}}dt} = -\frac{\rho_w d\overline{V}_{ma}}{\alpha_z dt} = -\frac{dW_{ma}}{\alpha_z dt} \tag{36}$$

where C is the constant of the material ($C = \rho_s\, s_s$) defined by Equations (22) and (23), $\delta V_{ma}$ the element of volume equal to $\delta V_{ma} = s_{ma}dz$, $\delta m_s = \rho_s\, s_s dz$, the mass of the solid phase concomitant, and $m_{s_{zma}}$ the mass of the solid phase at the level of the section at z associated with the volume variation $dV_{ma} = \frac{d(s_{ma})}{\alpha_z}$; $\alpha_z = C/m_{s_{zma}} \equiv L^{-1}$.

We give for $\alpha_z$ the following physical definition:

$$\frac{\rho_w s_{ma}dz}{Cdt} = -\frac{\rho_w \delta V_{ma}}{\frac{\delta m_s}{dz}dt} = -\frac{\rho_w dV_{ma}}{\alpha_z m_s dt} = -\frac{\rho_w d\overline{V}_{ma}}{\alpha_z dt} = -\frac{dW_{ma}}{\alpha_z dt} \tag{37}$$

where $\delta m_s$ is the solid mass element corresponding to $\delta V_{ma}$ and such that:

$$\frac{\delta m_s}{dz} = \alpha_z m_{sma} \tag{38}$$

$m_{sma}$ being the element (mass) of the solid phase associated with $dV_{ma}$ and such that:

$$\frac{dV_{ma}}{m_{sma}} = d\overline{V}_{ma} \tag{39}$$

Comparing Equations (39) and (40), we deduce that:

$$\alpha_z f_{ma} W_{ma} = \frac{d\, f_{ma}}{dz} W_{ma} = -\frac{dW_{ma}}{dt} \tag{40}$$

thus,

$$\alpha_z f_{ma} = \frac{d\, f_{ma}}{dz} \tag{41}$$

Since $f_{ma} = \frac{dz}{dt}$, we also have:

$$f_{ma} \frac{dW_{ma}}{dz} = \frac{dz}{dt}\frac{dW_{ma}}{dz} = \frac{dW_{ma}}{dt} \tag{42}$$

and from (44), the general equation:

$$\alpha_z W_{ma} f_{ma} = \frac{d f_{ma}}{dz} W_{ma} = -\frac{dW_{ma}}{dt} = -f_{ma}\frac{dW_{ma}}{dz} \tag{43}$$

The consequence of Equation (47) is that:

$$\frac{d\ln f_{ma}}{dz} = -\frac{d\ln W_{ma}}{dz} = \alpha_z \text{ and } \frac{d\ln(W_{ma}f_{ma})}{dz} = 0 \tag{44}$$

### 3.3.5. Flux Variables at Depth z

Just as we have defined the extensive variables at depth $z$ $\left(\overline{V}; W; W_{ma}; W_{mi}\right)$, we must also define the associated types of fluxes at depth $z$:

(1) The global or Eulerian flux ($fe_w$): the flux of water crossing the entire horizontal section ($s_t$), of the sample, without distinction of the surface actually crossed by this section:

$$fe_w = \frac{m_w}{\rho_w}\frac{dn_w}{s_t dt} = \frac{dV_w}{s_t dt} = \frac{s_w}{s_t}\frac{dz}{dt} = \frac{s_t}{s_t}\frac{dl_w}{dt} = \frac{dl_w}{dt} \tag{45}$$

where $m_w$ is the molecular mass of water and $n_w$ is the number of water molecules in the elementary water volume $dV_w = s_w dz$, which is in the elementary soil volume $dV_t = s_t dz$. This defines the elementary high of water ($dl_w$) such that $s_t dl_w = dV_w = s_w dz$. It follows that $\frac{dz}{dt}$ is the rate of transfer of water molecules through the surface ($s_t$), while $dl_w/dt$ is the rate of drainage of the height of water in the volume element ($dV = s_t dz$).

This implies the equivalence: $s_w dz = s_t dl_w$ and the different forms of writing of the Eulerian flux that we obtain by using Equation (28) of equivalences with the ratio of molecular surfaces:

$$fe_w = \frac{dl_w}{dt} = \frac{s_w}{s_t}\frac{dz}{dt} = \theta_w \frac{dz}{dt} = \frac{W}{\rho_w \overline{V}}\frac{dz}{dt}. \tag{46}$$

(2) The real flux, $f_w$, transfer speed of water molecules through their real surface of passage ($s_{ma}$), the surface occupied by the macro water molecules, the aqueous phase external to the primary aggregates:

$$f_w = \frac{m_w}{\rho_w}\frac{dn_w}{s_{ma}dt} = \frac{dV_w}{s_{ma}dt} = \frac{s_t}{s_{ma}}\frac{dl_w}{dt} = \frac{\rho_w \overline{V}}{W_{ma}}fe_w = \frac{fe_w}{\theta_{ma}} = \frac{W}{W_{ma}}\frac{dz}{dt} \tag{47}$$

where the ratio $\frac{dz}{dt}$ represents, as in Equation (29), the speed of movement of water molecules on the z axis.

(3) The molecular flux ($f_{ma}$), the real speed of the water molecules of the surface $s_{ma}$, is the speed (modulus) of the molecules of the macro phase determined by the chemical

potential of the phase different from $f_w$ related to the number of micro and macro water molecules leaving the surface during the time ($dt$):

$$f_{ma} = \frac{m_w}{\rho_w}\frac{dn_{wma}}{s_{ma}dt} = \frac{s_{ma}dz}{s_{ma}dt} = \frac{dz}{dt} \tag{48}$$

Indeed, the speed $\frac{dz}{dt}$ defined by this Equation (31) is that of the water molecules of the phase, namely, the speed of agitation of the molecules of the macro phase.

Thus, the Eulerian and real fluxes of the water in the pedostructure are linked to the speed of agitation of the molecules of the macro aqueous phase through the intermediary of the molecular flux $f_{ma}$ and therefore directly linked to the state variables of thermodynamic equilibrium of the phase (temperature, chemical potential, entropy etc.)

*3.4. Writing of the Physical Process at a z-Section Level of Scale*

Having defined the descriptive variables of the organization of the internal environment of the pedostructure at depth z, we can now introduce the physical processes that govern the movement of water (fundamental equation of dynamics) and the regulation of liquid phases by relative to the solid phase (thermodynamic equilibrium) due to evaporation of surface water.

3.4.1. Application of Newton's Law

The relation of flux with time, when it comes to a speed of movement, goes through the fundamental law of mechanics and Newton's 2nd law, mentioned above (14). These laws apply to the molecular flux ($f_{ma}$) which proceeds from the chemical potential gradient of the macro aqueous phase, the relationship of which is known with the water content of the phase at depth z:

The upward force $F_{ma}$, which drives the water molecules, of molecular mass: $m_w$, present at the $s_{ma}$ surface at the coordinate $z$ is equal to:

$$F_{ma} = m_w\gamma_{ma} = m_w\frac{df_{ma}}{dt} = +m_w\frac{d\mu}{dz} \tag{49}$$

They undergo an acceleration of:

$$\frac{df_{ma}}{dt} = \frac{d\mu_{va}}{dz} \tag{50}$$

The + sign of Equation (49) is negative in the literature but must be corrected as positive. In fact, the negative sign arises from the fact that the potential, $\mu$, is taken negative in a standard way, in accordance with current thinking about potentials. However, we showed [11] that the chemical potential of the thermodynamic phases ($\mu_{va}$ in this case) is always positive.

The products: $(f_{ma}W_{ma})$ and $\left(W_{ma}\frac{df_{ma}}{dt}\right)$ are, therefore, respectively: the linear momentum, $MLT^{-1}$ and the force of inertia, $MLT^{-2}$, of the $\bar{n}_{ma}$ molecules of mass $m_w$, both refer to the local structural mass $m_{s_z} = C/\alpha_z$ as we saw above.

Let us derive the linear momentum of molecules of water $(f_{ma}W_{ma})$ with respect to time:

$$\frac{d}{dt}(f_{ma}W_{ma}) = W_{ma}\frac{df_{ma}}{dt} + f_{ma}\frac{dW_{ma}}{dt} \tag{51}$$

By replacing $\frac{dW_{ma}}{dt}$ by its equivalent $\frac{-W_{ma}df_{ma}}{dz}$ given by Equation (34) and $\frac{df_{ma}}{dt}$ by $\frac{d\mu_{ma}}{dz}$ (50), we obtain:

$$\frac{d}{dt}(f_{ma}W_{ma}) = W_{ma}\frac{d\mu_{ma}}{dz} - f_{ma}W_{ma}\frac{df_{ma}}{dz} = W_{ma}\left(\frac{d\mu_{ma}}{dz} - \frac{df_{ma}^2}{2dz}\right) \tag{52}$$

We show in Appendix C that

$$\left( \frac{d\mu_{ma}}{dz} - \frac{d f_{ma}{}^2}{2dz} \right) = \alpha_t f_{ma} \tag{53}$$

where $\alpha_t$ is the constant of dimension *t*-1 of the time exponential in the relation:

$$W_{ma} = A_{t=0} \exp(\alpha_t t) + B \tag{54}$$

Thus, according to (52), we get the important relationship:

$$\frac{d}{dt}(f_{ma} W_{ma}) = \alpha_t f_{ma} W_{ma} \tag{55}$$

which leads to:

$$\frac{d \ln(W_{ma} \, f_{ma})}{dt} = \frac{d \ln f_{ma}}{dt} + \frac{d \ln W_{ma}}{dt} = \alpha_t = \frac{d}{dt} \ln(W_{ma} - B) \tag{56}$$

Furthermore, since $\frac{d(f_{ma} W_{ma})}{dz} = 0$ according to (48), the derivative of (52) with respect to $z$ is null:

$$\frac{d^2(f_{ma} W_{ma})}{dzdt} = \frac{d\alpha_t}{dz} W_{ma} f_{ma} + \alpha_t W_{ma} \frac{d f_{ma}}{dz} + \alpha_t f_{ma} \frac{dW_{ma}}{dz} = 0, \tag{57}$$

meaning that $\alpha_t$ is constant with depth.

3.4.2. Equations of $W_{ma}$ and $f_{ma}$ and Their Derivatives

We deduce from the above the equation of $W_{ma}$ depending on $t$ and $z$:

$$\begin{aligned} W_{ma} - W_{maf} &= W_{ma z=0} \exp(-\alpha_z z) - W_{maf} \\ &= \left( W_{ma^\circ} - W_{maf} \right) \exp(\alpha_t t) \exp(-\alpha_z z) \end{aligned} \tag{58}$$

where $B = W_{maf}$ and $A_{t=0} = \left( W_{ma^\circ} - W_{maf} \right)$ in Equation (54), $W_{maf}$ being the macro-water content of the soil surface at equilibrium with atmosphere, at the end of the experiment, and $W_{ma^\circ}$ being the macro-water content at saturation at the beginning of the experiment with the condition that $W_{maf} \leq W_{ma} \leq W_{ma^\circ}$.

Thus, we have:

$$\frac{dW_{ma}}{dz} = W_{ma z=0} \exp(-\alpha_z z) = -\alpha_z \left( W_{ma} - W_{maf} \right) \tag{59}$$

$$\frac{dW_{ma}}{dt} = \alpha_t \left( W_{ma^\circ} - W_{maf} \right) \exp(\alpha_t t) \exp(-\alpha_z z) = \alpha_t \left( W_{ma} - W_{maf} \right) \tag{60}$$

$$\frac{d^2 W_{ma}}{dzdt} = -\alpha_t \alpha_z \left( W_{ma^\circ} - W_{maf} \right) \exp(\alpha_t t) \exp(-\alpha_z z) = \frac{d^2 W_{ma}}{dtdz} \tag{61}$$

From (56) we can calculate the derivatives of $f_{ma}$ as functions of $W_{ma}$:

$$\frac{d \ln f_{ma}}{dt} = \alpha_t - \frac{dW_{ma}}{dt W_{ma}} = \alpha_t - \alpha_t \frac{(W_{ma} - B)}{W_{ma}} = \alpha_t \frac{B}{W_{ma}} \tag{62}$$

and, since $\frac{d \ln W_{ma}}{dt} = -\frac{d f_{ma}}{dz} = -\alpha_z f_{ma}$,

$$\frac{d f_{ma}}{dz} = \alpha_z f_{ma} = -\frac{d \ln W_{ma}}{dt} = \frac{d \ln f_{ma}}{dt} - \alpha_t = \alpha_t \left( \frac{B}{W_{ma}} - 1 \right), \tag{63}$$

which gives:

$$f_{ma} = \frac{\alpha_t}{\alpha_z} \frac{(B - W_{ma})}{W_{ma}} = \frac{-dW_{ma}/dt}{-dW_{ma}/dz} = \frac{dz}{dt} \tag{64}$$

Finally, retaking (62) and using (64), we obtain:

$$\frac{df_{ma}}{dt} = \alpha_t \frac{B}{W_{ma}} f_{ma} = \frac{\alpha_t^2}{\alpha_z} \frac{B}{W_{ma}} \left( \frac{B - W_{ma}}{W_{ma}} \right) \tag{65}$$

3.4.3. Application of the Equilibrium Equations between the Two Pedostructure Aqueous Phases

The pressure balance between the two aqueous phases which is established simultaneously with the migration of macro water to the surface can be seen as follows.

The molecular flux $f_{ma}$ of the macro phase is really the molecular speed of the molecules of this phase when they pass through the section $\bar{s}_{ma}$ under the effect of a potential gradient, specific for this phase, determined at $z$ by the equation:

$$\frac{dH}{dz} = \frac{dh_{ma}}{dz} - \rho_w g = \rho_w \left( \frac{d\mu_{ma}}{dz} - g \right) \tag{66}$$

The molecules of the micro phase (of potential $\mu_{mi}$) that are found in the clay plasma of the primary aggregates are themselves subjected to the pressure difference $(h_{ma} - h_{mi})$, which appears between the two phases as soon as there is a change in macro water content $W_{ma}$ at z, i.e., as soon as a variation in the flux of molecules of this phase along the z axis appears $(df_{ma}/dz \neq 0)$.

In this case, $(df_{ma}/dz \neq 0)$, the pressure balance between the two phases expressed by $h_{ma} = h_{mi}$ is broken and must be re-established by a lateral flux of molecules from the micro phase to the macro phase.

We can then describe the process of water migration in the sample following the evaporation of water at the surface as follows: the variation of the flux of inter-aggregate water at depth $z$, $\frac{df_{ma}}{dz} \neq 0$, has the effect of a change in water content $W_z$ at this same depth z, which simultaneously causes a new micro to macro flux totally determined by the equilibrium pressure equation $h_{ma} = h_{mi}$. This equality was studied above, giving rise to relations (10) to (12).

Moreover, starting from the equilibrium condition: $h_{mi} = h_{ma}$, we have every moment

$$\mu_{mi}^{eq} - \mu_{miSat} = \mu_{ma}^{eq} - \mu_{maSat} \tag{67}$$

which is written, according to the local values of $W_{mi}$ and $W_{ma}$:

$$\frac{\overline{E}_{mi}}{W_{mi}} - \frac{\overline{E}_{mi}}{W_{miSat}} = \frac{\overline{E}_{ma}}{W_{ma}} - \frac{\overline{E}_{ma}}{W_{maSat}} \tag{68}$$

By setting the constant parameters of the shrinkage curve [6]:

$$A = (\mu_{maSat} - \mu_{miSat}) = \frac{\overline{E}_{ma}}{W_{maSat}} - \frac{\overline{E}_{mi}}{W_{miSat}} \text{ and } \overline{E} = \overline{E}_{mi} + \overline{E}_{ma} \tag{69}$$

we get the following equalities:

$$\frac{W_{mi}}{\overline{E}_{mi}} = \frac{W_{ma}}{\overline{E}_{ma} - AW_{ma}} = \frac{W}{\overline{E} - AW_{ma}}; \tag{70}$$

showing that the ratios $\frac{W_{mi}}{W_{ma}}$, $\frac{W}{W_{mi}}$ and $\frac{W}{W_{ma}}$ are all functions of $W_{ma}$ alone. Therefore, we have:

$$W = \frac{W_{ma}(\overline{E} - AW_{ma})}{(\overline{E}_{ma} - AW_{ma})} \text{ et } W_{mi} = \frac{\overline{E}_{mi}W_{ma}}{\overline{E}_{ma} - AW_{ma}} \tag{71}$$

Having the distribution of $W_{ma}$ in space and its variation with time, we automatically have the values and variations of $W$ and $W_{mi}$ in any point of the medium.

The following equations can be verified:

$$\frac{dW_{mi}}{dW_{ma}} = \frac{\overline{E}_{ma}}{\overline{E}_{mi}} \frac{\overline{E}_{mi}2}{\left(\overline{E}_{ma} - AW_{ma}\right)^2} = \frac{\overline{E}_{ma}}{\overline{E}_{mi}}\left(\frac{W_{mi}}{W_{ma}}\right)^2 \tag{72}$$

leading to:

$$\frac{dW}{dt} = \frac{dW_{ma}}{dt}\left(1 + \frac{\overline{E}_{ma}}{\overline{E}_{mi}}\left(\frac{W_{mi}}{W_{ma}}\right)^2\right) = \alpha_t\left(W_{ma} - W_{maf}\right)\left(1 + \frac{\overline{E}_{ma}}{\overline{E}_{mi}}\left(\frac{W_{mi}}{W_{ma}}\right)^2\right) \tag{73}$$

and

$$\frac{dW}{dz} = \frac{dW_{ma}}{dz}\left(1 + \frac{\overline{E}_{ma}}{\overline{E}_{mi}}\left(\frac{W_{mi}}{W_{ma}}\right)^2\right) = -\alpha_z W_{ma}\left(1 + \frac{\overline{E}_{ma}}{\overline{E}_{mi}}\left(\frac{W_{mi}}{W_{ma}}\right)^2\right) \tag{74}$$

The continuity Equation (39) becomes:

$$\frac{dW}{dt} = -\rho_w \overline{V}\frac{df e_w}{dz} = -W_{ma}\frac{df_w}{dz} = -W_{ma}\frac{df_{ma}}{dz}\left(1 + \frac{\overline{E}_{ma}}{\overline{E}_{mi}}\left(\frac{W_{mi}}{W_{ma}}\right)^2\right) \tag{75}$$

## 4. Results

### 4.1. Linear Relationships between Wz, Wt and Time

The characteristic retention curve of the sample $h(W)$ is shown in Figure 5, which also shows the two curves ($h_1$ and $h_2$) of the continuous reading of tensiometers T1 and T2 as a function of the total water content ($W_t$) of the sample. The curves are homothetic: the total water content of the sample ($W_t = (M - M_s)/M_s$) corresponds to the values of the suction pressures $h_1$ and $h_2$ measured by the tensiometers and to the local water contents $W_1$ and $W_2$ that can be read on the retention curve $h(W)$, characteristic of the sample.

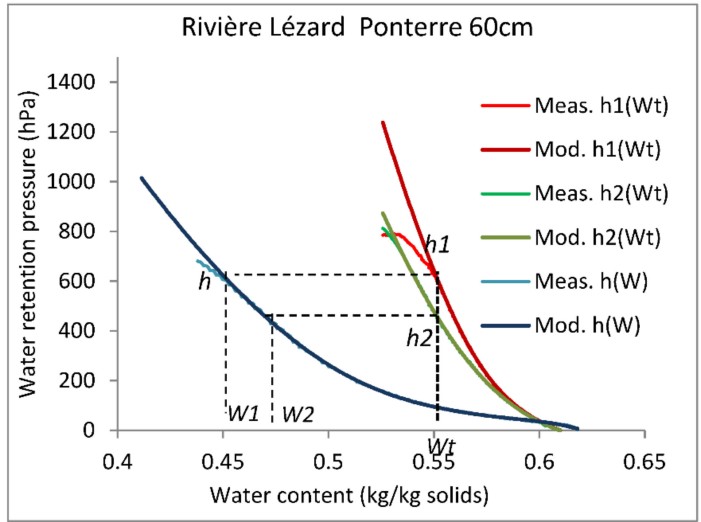

**Figure 5.** Representation on the same graph of the water retention characteristic curve $h(W)$, measured and modelled, and of the tensiometric reading $h_1$ and $h_2$ according to the total water content $W_t$. Modelled curves of $h(W)$, $h_1(W_t)$ and $h_2(W_t)$ used the same pedostructural parameters characteristic of the sample: $W_{maSat}$, $W_{miSat}$, $\overline{E}_{ma}$ and $\overline{E}_{mi}$.

The following relationships are observed:

$$W_{z1} = a_1 W_t + b_1 \text{ and } W_{z2} = a_2 W_t + b_2 \tag{76}$$

where $a_i$ and $b_i$ are constants associated with the depth $z_i$. At a given depth, the local water content is in constant proportion to the overall water content of the sample. The same applies to the difference in water content between two depths:

$$W_{z2} - W_{z1} = W_t(a_2 - a_1) - (b_2 - b_1) \tag{77}$$

Let us find the relationship between $a_i$ and $b_i$. At water saturation $W_0$ of the sample,

$$W_{zsat} = a_i W_{tsat} + b_i \text{ and } W_{tsat} = W_{zsat} = W_0 \tag{78}$$

thus,

$$b_i = W_0(1 - a_i) \tag{79}$$

and

$$a_i = \frac{W_z - W_0}{W_t - W_0} \tag{80}$$

For $a_i$ to be time independent, whatever z, it is necessary that:

$$\frac{da_i}{dt} = -\frac{dW_t}{dt}\frac{W_z - W_0}{(W_t - W_0)^2} + \frac{dW_z}{dt}\frac{W_t - W_0}{(W_t - W_0)^2} = 0 \tag{81}$$

$$\frac{dW_t}{dt}(W_z - W_0) = \frac{dW_z}{dt}(W_t - W_0) \tag{82}$$

or else

$$\frac{d\ln(W_t - W_0)}{dt} = \frac{d\ln(W_z - W_0)}{dt} \tag{83}$$

The relation between the water content local, $W_Z$, and total, $W_t$, is such that:

$$\frac{dW_z}{dt} \Big/ \frac{dW_t}{dt} = \frac{(W_z - W_0)}{W_t - W_0} = a_i \tag{84}$$

$W_z$ being defined as the ratio of the areas $s_w = s_{ma} + s_{mi}$ and $s_s$ at z: $W_z = \left(\frac{s_{ma} + s_{mi}}{s_s}\right)_z$.

The 2 graphs in Figure 6 give the values of $a_i$ and of $\frac{dW_z}{dt}$ at the two positions of the tensiometers $z_1$ and $z_2$, which gives us, according to (83), $\frac{dW_t}{dt} = \frac{dW_z/dt}{a_i} = -8.56 \times 10^{-5}$ min$^{-1}$ and $-8.57 \times 10^{-5}$ min$^{-1}$, respectively.

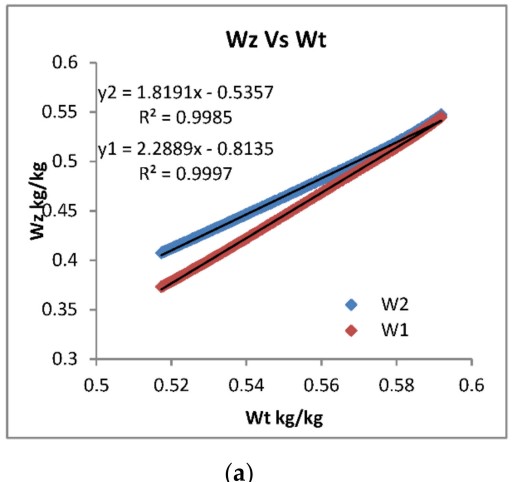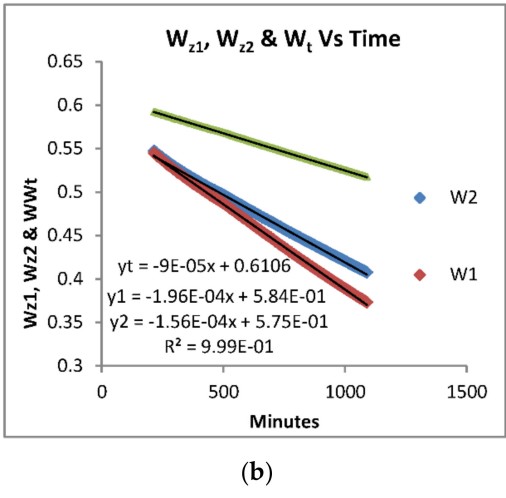

(**a**)          (**b**)

**Figure 6.** Linear relationships between *Wz*, *Wt* and *Time*. (**a**) represents the local water content at depths *z1* and *z2* according to the global water content of the sample; (**b**) represents the time dependence of the two local water contents (at depths *z1* and *z2*) and the global water content. The z-area ratio $a_i$ can be read on the figure (**a**): 2.29 for *z* = *z1* and 1.82 for *z* = *z2*.

### 4.2. Logarithmic Relation between $W_z$ and $W_{ma}$-$W_{maf}$

We can see in Figure 7 that the relation between $W_z$ and $W_{ma}$ is, for the two cases of z, a simple logarithmic function such as:

$$\frac{W_z}{W_c} = \ln\left(W_{maz} - W_{maf}\right) + C \tag{85}$$

where $C$ is a dimensionless constant and $1/W_c = \alpha_w$ is a constant parameter of the exponential of $W_z$.

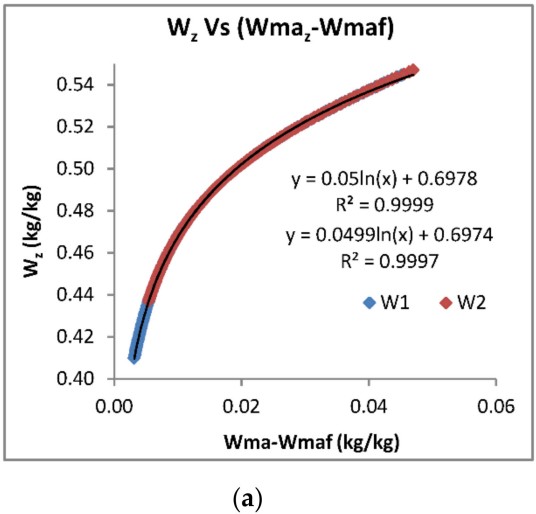

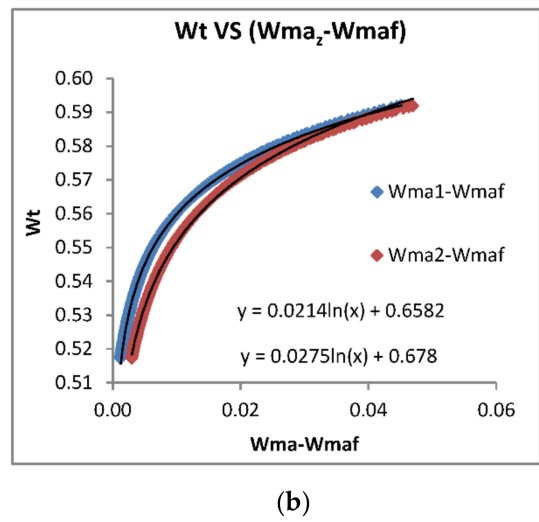

(**a**)             (**b**)

**Figure 7.** Relationship between $W_t$, $W_z$ and $\left(W_{maz} - W_{maf}\right)$. Rivière Lézarde Ponterre (halloysite soil) 60–65 cm. (**a**) represents the two local water contents ($W_z$ and $W_{maz}$) relationship at two given depths (**b**) represents the global water content of the sample as a function of the local macro water content at a given depth $z$. The constant $W_c$ in Equation (85) is read on the left figure is 0.04995 and 0.0500 kg of water/kg of soil.

What is remarkable is that this logarithmic form of $W_z$ (84) exactly represents the Equation (71) of $W_z$ function of $W_{maz}$:

$$\frac{W_z}{W_c} = \ln\left(W_{maz} - W_{maf}\right) + C = \frac{1}{W_c} \frac{W_{ma}\left(\overline{E} - AW_{ma}\right)}{\left(\overline{E}_{ma} - AW_{ma}\right)} \tag{86}$$

By differentiating (84) with respect to time and using the relation (60) giving $\frac{dW_{ma}}{dt}$, we obtain:

$$\frac{\left(W_{ma} - W_{maf}\right)}{W_c} \frac{dW_z}{dt} = \frac{dW_{ma}}{dt} = \alpha_t \left(W_{ma} - W_{maf}\right) \tag{87}$$

We, therefore, have whatever z in the unsaturated zone:

$$\frac{dW_z}{dt} = \alpha_t W_c = cte \tag{88}$$

and according to the relation (73) that exists between $\frac{dW_z}{dt}$ and $\frac{dW_{ma}}{dt}$:

$$W_c = \left(W_{ma} - W_{maf}\right)\left(1 + \frac{\overline{E}_{ma}}{\overline{E}_{mi}}\left(\frac{W_{mi}}{W_{ma}}\right)^2\right) = \left(W_{ma} - W_{maf}\right)R_z \tag{89}$$

### 4.3. The Fundamental Relationships between Flux, Water Potential and Water Content at Macroscopic Scale

#### 4.3.1. Central Role of $W_{ma}$

Figure 8 shows the experimental result of the relationship between the macro water content gradients $\frac{\Delta W_{ma}}{\Delta z}$ and the pressure $\frac{\Delta h}{\Delta z}$ of a thin soil layer and the average water content of this layer.

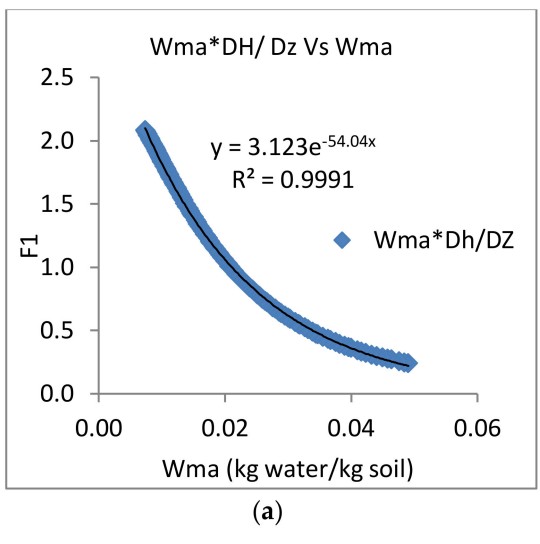
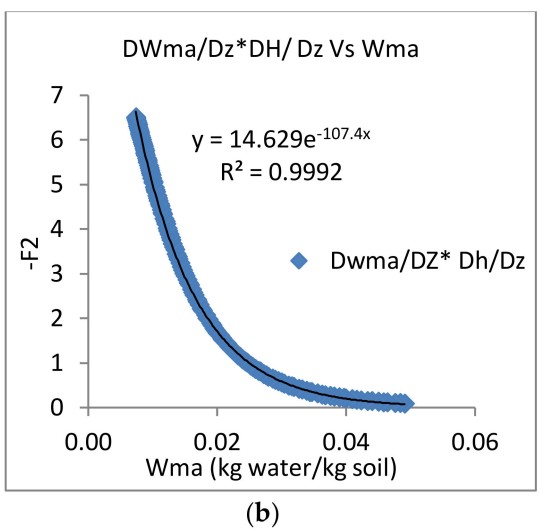

(a)        (b)

**Figure 8.** Measurement result of functions F1 and F2 defined hereafter in the text, which are simple exponentials of $W_{ma}$: (**a**) represents F1 as function of $W_{ma}$ and (**b**) F2 as function of $W_{ma}$.

We are in the case of a systemic discretization of the soil medium to apply transfer equations [9]. The values of $W_{ma1}$ and $W_{ma2}$ are calculated from the data of tensiometers T1 and T2 using the characteristic soil retention curve. They are supposed to represent the average water contents in the 1 cm thick slices around tensiometers 1 and 2. The products $W_{ma}\frac{\Delta h}{\rho_w \Delta z}$, $\frac{\Delta h}{\rho_w \Delta z}\frac{\Delta W_{ma}}{\Delta z}$ and $\frac{\Delta \ln W_{ma}}{\Delta z}$ named F1, F2 and F3 are presented as functions of $W_{ma(1,2)}$, mean of $W_{ma}$ between z1 and z2.

We observe three simple exponentials of $W_{ma}$, two of which are represented in Figure 8:

$$F1 \equiv W_{ma}\frac{\Delta H}{\rho_w \Delta z} = k_1 \exp(\alpha_1 W_{ma}); \ k_1 = \ 3.123 \ \mathrm{Jm^{-1}} \ \mathrm{and} \ \alpha_1 \ = \ -54.04 \ \mathrm{kg\_soil/kg\_water} \tag{90}$$

$$F2 \equiv \frac{\Delta W_{ma}}{\rho_w \Delta z}\frac{\Delta H}{\Delta z} = k_2 \exp(\alpha_2 W_{ma}); \ k_2 = \ -14.629 \ \mathrm{Jm^{-2}} \ \mathrm{and} \ \alpha_2 \ = \ -107.4 \ \mathrm{kg\_soil/kg\_water} \tag{91}$$

$$F3 \equiv \frac{\Delta \ln W_{ma}}{\Delta z} = k_3 \exp(\alpha_3 W_{ma}); \ k_3 = \ -4.721 \ \mathrm{m^{-1}} \ \mathrm{and} \ \alpha_3 \ = \ -53.61 \ \mathrm{kg\_soil/kg\_water} \tag{92}$$

To interpret these results, recall that the basic variables, suction pressure (*h*), chemical potential ($\mu_{ma}$), and molecular flux of the mobile phase ($f_{ma}$), are linked by the relation:

$$\frac{dh}{\rho_w dz} = \frac{d\mu_{ma}}{dz} = \frac{d\mu_{mi}}{dz} = \frac{df_{ma}}{dt} \tag{93}$$

and knowing that

$$d\ln\mu_{ma} = -d\ln W_{ma} \ \mathrm{et} \ W_{ma}\mu_{ma} = \overline{E}_{ma} \tag{94}$$

we then have the following equalities:

$$W_{ma}\frac{dh}{\rho_w dz} = W_{ma}\frac{d\mu_{ma}}{dz} = W_{ma}\frac{df_{ma}}{dt} = \overline{E}_{ma}\frac{d\ln\mu_{ma}}{dz} = -\overline{E}_{ma}\frac{d\ln W_{ma}}{dz} \tag{95}$$

Assuming that the discretization is fine enough to maintain at the macroscopic scale the relationship observed at the molecular scale between extensive and intensive variables, we should observe, after (89)–(91) and (94): $F1 = -\overline{E}_{ma}F3$ and $F1F3 = F2$

$$F1F3 = F2 \equiv \frac{\Delta W_{ma}}{\Delta z}\frac{\Delta h}{\rho_w \Delta z} = -\frac{k_1{}^2}{\overline{E}_{ma}}\exp(2\alpha_1 W_{ma}) \tag{96}$$

$$F3 \equiv \frac{\Delta \ln W_{ma}}{\Delta z} = -\frac{k_1}{\overline{E}_{ma}}\exp(\alpha_1 W_{ma}) \tag{97}$$

We can notice the good accordance between the measured values of parameters and the theoretical relationships between them

$$k_2 = -\frac{k_1{}^2}{\overline{E}_{ma}} = k_1 k_3 \text{ and } \alpha_1 = \alpha_3 = \alpha_2/2 \tag{98}$$

### 4.3.2. Pedostructure Water Conductivity $k_{ps}$

Thus, we have all the physical equations determining the space–time relationship of variation of the three variables describing the dynamics of the medium: the fluxes, water contents and chemical potentials of the two aqueous phases.

Recall the equation of continuity (74) that takes account of the thermodynamic equilibrium. Using the relation $f_{ma} = \frac{dz}{dt}$, we can write:

$$f_{ma}\frac{dW}{dz} = \frac{dz}{dt}\frac{dW}{dz} = \frac{dW}{dt} \tag{99}$$

so the equation of continuity can be written such as:

$$\frac{dW}{dt} = \frac{dW_{ma}}{dt}\left(1 + \frac{\overline{E}_{mi}}{\overline{E}_{ma}}\left(\frac{\mu_{ma}}{\mu_{mi}}\right)^2\right) = -W_{ma}\frac{df_w}{dz} = f_{ma}\frac{dW}{dz} = -\rho_w \overline{V}\frac{df e_w}{dz} \tag{100}$$

Furthermore, experience has shown that

$$\frac{dW}{dt} = \alpha_t W_c = cte \tag{101}$$

$W_{ma}$ being an exponential of time and space, we deduce from the fact that $W_{ma}\frac{df_w}{dz} = -\frac{dW}{dt} = cte$ (Equations (98) and (99)) that $\frac{df_w}{dz}$ and therefore also $f_w$ are simple exponentials with the same coefficients as $W_{ma}$. Thus, as the soil water conductivity by definition is written:

$$k_{ps} = \frac{f e_w}{dh/dz} = \frac{f_w}{dh/dz}\frac{f e_w}{f_w} = \frac{f_w}{dh/dz}\theta_{ma}, \tag{102}$$

by multiplying (101) up and down by $\alpha_z W_{ma} = -\frac{dW_{ma}}{dz}$ (47) we get:

$$k_{ps} = \frac{\alpha_z W_{ma}f_w}{\alpha_z W_{ma}dh/dz}\theta_{ma} = \frac{W_{ma}(df_w/dz)}{(W_{ma}/dz)(dh/dz)}\theta_{ma} = \frac{-dW/dt}{F2}\theta_{ma} \tag{103}$$

and using Equation (100):

$$k_{ps} = \frac{-\alpha_t W_c}{k_2}\theta_{ma}\exp(-\alpha_2 W_{ma}) \tag{104}$$

The constants: $\alpha_t$, $W_c$, $k_2 = -\frac{k_1{}^2}{\overline{E}_{ma}}$ and $\alpha_2 = 2\alpha_1$, are all determined by measurement as we showed above (97).

## 5. Discussion

The systemic modeling of the hydrostructural soil water properties by the model Kamel [9,12] already accounted for the levels of internal organization of the "soil factory". It precisely identified the pedostructure as assembly of primary peds containing two thermodynamically distinct aqueous phases, intra-aggregate (macro) phase and inter-aggregate (micro) phase. However, this modeling still ignored the lower levels of organization (molecular and atomic) of the fluid phases of the pedostructure. It retained, therefore, a semi-empirical character because it is at these two levels of organization that the variables of temperature, pressure, entropy and chemical potential have their basis of definition, as our previous study [11] showed. Furthermore, all these variables intervene in the hydro-thermodynamic equilibria of the soil at higher levels of organization (soil suction gradient, soil water retention curve, hydric conductivity, etc.). By understanding the internal organization of the aqueous phases and their roles in the evaporation process, we were able to relate the variables at each level to each the others in a comprehensive and orderly manner.

The present study focused on the notions of flux: molecular flux ($f_{ma}$), water flux ($f_w$, $f_{ma}$ and $f_{mi,}$) and Eulerian flux ($f_{ew}$). The $f_{ma,}$ flux is said to be molecular because it is equal to the celerity of the molecules of the phase and, therefore, linked to the chemical potential of the phase as we have shown it. Application of the Newton's second law makes it possible to identify the gradient of the chemical potential of the inter-aggregates aqueous phase (macro) to the time derivative of its molecular flux, leading to the relation: $\frac{d\mu_{ma}}{dz} = \frac{df_{ma}}{dt}$, which has the dimensions of an acceleration ($LT^{-2}$). This important relationship could not be obtained without the acknowledgement of these molecular and atomic organization levels. This allows us to say that we have solved the Navier–Stokes equation for the particular case of the water flux in pedostructure during its drying by evaporation at its upper surface.

Above this molecular level, there are the nested levels of organization that we have already dealt with exhaustively in hydrostructural pedology [1,2]. The present study has defined and highlighted the junction point of both worlds by studying the molecular and non-molecular descriptive variables (intensive and extensive) attached to the z-depth where each descriptive variable can be considered punctual and continuous.

Finally, the equation of water conductivity (103) is given here theoretically for the first time from a combination of hydro-thermodynamic variables. It has the same form as the one we found semi-empirically and published recently in [13]. This result solves one of the many points of soil science questioned [14] and which was not yet fully resolved by hydrostructural pedology; in particular, the physical equation of the water conductivity of unsaturated soils and its measurement [15]. The resolution of this last point confirms the validity of the "systemic approach" such as it was redefined by the authors [1–3] to face the "black-box" modelling in soil hydrodynamics and thermodynamics. It also justifies the new methods of characterization of the water functions of the soil [16–18].

## 6. Conclusions

A long-standing problem in fundamental physics has been solved and experimentally confirmed in this article: that of the dynamics of water in unsaturated soil. The exact physical equation for non-saturated hydric conductivity has been found; this by reconsidering the equations of Euler and Richards with a new approach, systemic and molecular, of thermodynamics. The methodology for obtaining parameters of the pedostructure hydric conductivity curve of soil is also given in the article, valid for all types of soil, swelling or not. At the same time, we discovered that the link between the two domains of the soil science: soil hydrodynamics and soil water thermodynamics is provided by Newton's 2nd law, which links the acceleration of molecules of the macro aqueous phase to the gradient of the chemical potential of this phase.

These results confirm the validity of two recent theories in environmental science thanks to which they were obtained: the systemic approach which gave birth to hydrostructural pedology and the systemic and molecular thermodynamics of soil water.

**Author Contributions:** Conceptualization, writing—original draft preparation, E.B.; writing—review and editing, R.H.M. All authors have read and agreed to the published version of the manuscript.

**Funding:** This research received no external funding. Field sampling and laboratory data were acquired in 2003–2005 in Martinique by the first author in the frame of its research work at IRD.

**Institutional Review Board Statement:** Not applicable.

**Informed Consent Statement:** Not applicable.

**Data Availability Statement:** The data presented in this study are available on request from the corresponding author. The data are not publicly available due to non-prepared presentation.

**Acknowledgments:** The authors warmly thank Hassan Boukcim (agro-biologist, CEO of Valorhiz) and Amjad T. Assi, research assistant professor at Texas A&M University for their long and constant support for the applied aspects of this work.

**Conflicts of Interest:** The authors declare no conflict of interest.

## Appendix A

*Appendix A.1. The Richards Equation, an Eulerian Point of View*

The Richards equation is the combination of two physical equations of different conceptual origins:

(a) conservation law of the matter which is expressed mathematically by the Euler equation

$$\frac{d\theta_w}{dt} = -\frac{dfe_w}{dz},\tag{A1}$$

where $\theta_w$ is the volume water content of the soil and where $fe_w$ is the Eulerian flow, understood as the volume flow velocity on $z$ (upward positive);

(b) Darcy′s law extended to unsaturated porous media, the classical expression of which is

$$fe_w = k\Delta H/\Delta z\tag{A2}$$

where $\frac{\Delta H}{\Delta z} = \frac{\Delta h}{\Delta z} - \rho_w g \frac{\Delta z}{\Delta z}$ is the pressure gradient of the soil water with respect to z, sum of the water retention pressure gradients and the pressure gravity.

This relationship is called "Darcy′s law extended to unsaturated soils" and its application to the Euler equation constitutes the Richards equation used in all soil-water models of water dynamics in soil:

$$\frac{d\theta_w}{dt} = -\frac{\partial}{\partial z}K\left(\frac{dH}{dz}\right)\tag{A3}$$

The question that arises when one goes from the verified Darcy law for saturated soils to the extension of the law for unsaturated soils concerns the surface area of the flux at depth z. We generally take $s_t$ the total surface of the sample but would it not be $s_w$, the surface occupied by the water molecules, which decreases with the water content, or else only $s_{ma}$, the surface corresponding to the inter-aggregate of water, which should be considered? The systems approach helps answer this question as we will see below. In fact, the variable $\theta_w$, which is the ratio of the volume of water to the volume of soil $\theta_w = V_w/V_t$, is a non-systemic variable defined according to the notion of REV (Representative Elementary Volume), which implies that the variable $fe_w$ cannot be defined by Equation (8): we do not know what it is spatially represents. The notions of surfaces occupied by water molecules and solids at the depth z of the cylinder and of common thickness of the soil layer do not appear in the concept of REV.

The so-called Richards Equation (A3) is therefore empirical and cannot be related to fundamental physics. To write it correctly, we must first understand the exact physical

meaning of the flux variables above, $f_w$ et $fe_w$, but also that of the thermodynamic variables that describe the energy equilibria that are established at the molecular scale between the molecules of each fluid phase as well as at the global scale between the thermodynamic phases of the pedostructure. Indeed, these thermodynamic variables, like chemical potential, temperature and entropy, take on their full physical significance at the molecular scale as we have shown in [11]. We have to take this level of molecular scale into account to describe the process of migration of water molecules in the soil (its pedostructure) submitted to a potential gradient at thermodynamic equilibrium.

*Appendix A.2. Euler's Equation for Conservation of the Mass*

As mentioned above, the Richards equation comes from the introduction of Darcy's law "extended to unsaturated media" into the Euler continuity equation. We need to study the terms, including the meaning of flow, to relate it exactly to the speed of molecules and the thermodynamic variables that were defined at the molecular level in the first part.

Euler's equation for the conservation of the mass of water moving in the soil is written:

$$\frac{\partial \rho}{\partial t} + \nabla(\rho_w v) = 0 \tag{A4}$$

where $\rho$ is the volume concentration of water in the soil (mass of water in the volume $V$ of soil: $\rho = \frac{M_w}{V}$), $t$ is time and $v$ the Eulerian velocity of the fluid (water) with density $\rho_w$ assumed constant. The Eulerian speed can be identified, except for the sign, with a flow that we will call Eulerian flux and write: $fe_w$ ($\rho_w = \frac{M_w}{V_w}$, $v = fe_w = \pm \frac{dl_w}{dt}$).

Let us rewrite this Equation (A4) using the systemic variables ($W$ and $\overline{V} = V/m_s$) rather than the non-systemic variables ($\rho$ and $\theta = V_w/V$) in order to discern the internal process variables involved in the migration of water in the sample subjected to evaporation. Let us first recall the existing relationships between these types of variables, $m_w$ and $m_s$ being the mass of water and solids contained in the volume $V$ of soil (pedostructure):

$$\rho = \frac{m_w}{V} = \rho_w \frac{V_w}{V} = \rho_w \theta = \frac{m_w/m_s}{V/m_s} = \frac{W}{\overline{V}} = \frac{\rho_w \overline{V}_w}{\overline{V}} \tag{A5}$$

$$\frac{d\rho}{dt} = \frac{d(W/\overline{V})}{dt} = \frac{\rho_w d\theta}{dt} \tag{A6}$$

since $\rho_w = cte$ at constant temperature and pressure.

The second term is written such as:

$$\nabla(\rho_w v) = \rho_w \nabla v = \rho_w \left( \frac{dfe_w}{dz} + \frac{dfe_w}{dy} + \frac{dfe_w}{dx} \right) \tag{A7}$$

where $fe_w = v$ is the Eulerian flow which has the dimensions of a velocity, in LT-1.

Thus, the Euler equation that is known in hydrology is:

$$\frac{d\theta}{dt} = -\frac{dfe_w}{dz}$$

## Appendix B

*Appendix B.1. Equations of $W$, $f_w$ and Their Derivatives According to $W_{ma}$ and $f_{ma}$*

Consider now the products $f_w W$ and $fe_w \rho_w \overline{V}$; we have, according to the definition of the derivatives of fluxes (39–41):

$$\frac{df_w}{dz} = -\frac{1}{W_{ma}} \frac{dW}{dt} \tag{A8}$$

and

$$\frac{df e_w}{dz} = -\frac{1}{\rho_w \overline{V}} \frac{dW}{dt} \tag{A9}$$

We then obtain the general equation, similar to the Euler equation written with the systemic variables:

$$\frac{dW}{dt} = -\rho_w \overline{V} \frac{df e_w}{dz} = -W_{ma} \frac{df_w}{dz} \tag{A10}$$

Moreover, using the relation $f_{ma} = \frac{dz}{dt}$ we can write:

$$f_{ma} \frac{dW}{dz} = \frac{dz}{dt} \frac{dW}{dz} = \frac{dW}{dt} \tag{A11}$$

so we have

$$-W_{ma} \frac{df_w}{dz} = f_{ma} \frac{dW}{dz} = \frac{dW}{dt} \tag{A12}$$

that we can compare to

$$-W_{ma} \frac{df_{ma}}{dz} = f_{ma} \frac{dW_{ma}}{dz} = \frac{dW_{ma}}{dt} \tag{A13}$$

By subtracting the two equations term by term, we obtain:

$$-W_{ma} \left( \frac{df_w}{dz} - \frac{df_{ma}}{dz} \right) = f_{ma} \left( \frac{dW}{dz} - \frac{dW_{ma}}{dz} \right) = \frac{dW}{dt} - \frac{dW_{ma}}{dt} \tag{A14}$$

equivalent to:

$$-W_{ma} \left( \frac{df_w}{dz} - \frac{df_{ma}}{dz} \right) = f_{ma} \left( \frac{dW_{mi}}{dz} \right) = \frac{dW_{mi}}{dt} \tag{A15}$$

Defining $f_{mi} = f_w - f_{ma}$ as the virtual speed of molecules of the micro phase at z, we have the following fundamental relationships:

$$-W_{ma} \frac{df_{mi}}{dz} = f_{ma} \frac{dW_{mi}}{dz} = \frac{dW_{mi}}{dt} \tag{A16}$$

We find here the central role of $f_{ma}$ and $W_{ma}$:

$$f_{ma} = \frac{dW_{mi}/dt}{dW_{mi}/dz} = \frac{dW_{ma}/dt}{dW_{ma}/dz} = \frac{dW/dt}{dW/dz} \tag{A17}$$

and

$$W_{ma} = -\frac{dW_{mi}/dt}{df_{mi}/dz} = -\frac{dW_{ma}/dt}{df_{ma}/dz} = -\frac{dW/dt}{df_w/dz} \tag{A18}$$

### Appendix C

*Appendix C.1. Application of Newton's Law, $\frac{d(f_{ma}W_{ma})}{dt} = \alpha_t f_{ma} W_{ma}$ Demonstration*

Let us try to determine the relation between the two terms $\frac{d\mu_{ma}}{dz}$ and $\frac{df_{ma}2}{2dz}$ of relation (52). The product $f_{ma}W_{ma}$ written with its fundamental variables is equal to $-\frac{dW_{ma}}{\alpha_z dt}$ from Equation (40). The derivative with respect to time is, therefore, the second derivative of $W_{ma}$:

$$\frac{d(f_{ma}W_{ma})}{dt} = -\frac{d^2 W_{ma}}{\alpha_z dt2} \tag{A19}$$

From Equations (52) and (A19) we therefore have:

$$\frac{d^2 W_{ma}}{\alpha_z dt^2} = \frac{d(f_{ma}W_{ma})}{dt} = -\frac{df_{ma}}{dt} W_{ma} - \frac{dW_{ma}}{dt} f_{ma} = -W_{ma} \left( \frac{d\mu_{ma}}{dz} - \frac{df_{ma}^2}{2dz} \right) \tag{A20}$$

Suppose that $\frac{dW_{ma}}{dt}$ is an exponential function of time, in its most general form, as will be verified experimentally:

$$W_{ma} = A_{t=0}\exp(\alpha_t t) + B \text{ and } \frac{d2W_{ma}}{dt2} = \alpha_t{}^2 A_{t=0}\exp(\alpha_t t) = \alpha_t \frac{dW_{ma}}{dt} = \alpha_t{}^2(W_{ma} - B) \tag{A21}$$

Equation (A20) is then written:

$$\frac{\mathrm{d}^2 W_{ma}}{\alpha_z dt2} + \frac{dW_{ma}}{dt} f_{ma} = -\frac{df_{ma}}{dt} W_{ma} \tag{A22}$$

which, according to (55) and (50), can be put in the form:

$$\frac{dW_{ma}}{dt}\left(\frac{\alpha_t}{\alpha_z} + f_{ma}\right) = -\frac{df_{ma}}{dt} W_{ma} = -\frac{d\mu_{ma}}{dz} W_{ma} \tag{A23}$$

By dividing all the members of Equation (A23) by $W_{ma}$ and using Equation (34), we obtain the relation which links together $\frac{d\mu_{ma}}{dz}$, $\frac{df_{ma}}{dt}$ and $\frac{df_{ma}}{dz}$:

$$\frac{df_{ma}}{dt} = \frac{d\mu_{ma}}{dz} = -\frac{d\ln W_{ma}}{dt}\left(\frac{\alpha_t}{\alpha_z} + f_{ma}\right) = \frac{df_{ma}}{dz}\left(\frac{\alpha_t}{\alpha_z} + f_{ma}\right) \tag{A24}$$

Using the relation $\frac{df_{ma}}{dz} = \alpha_z f_{ma}$ (45), (A24) becomes:

$$\frac{df_{ma}}{dt} = \frac{d\mu_{ma}}{dz} = \alpha_t f_{ma} + \frac{1}{2}\frac{df_{ma}{}^2}{dz} \tag{A25}$$

From where we get

$$\frac{d\mu_{ma}}{dz} - \frac{1}{2}\frac{df_{ma}{}^2}{dz} = \alpha_t f_{ma} \tag{A26}$$

which, reported in Equation (A20), gives:

$$\frac{d}{dt}(f_{ma}W_{ma}) = \alpha_t W_{ma} f_{ma} \tag{A27}$$

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
