# Peer review of "Hydrostructural Pedology, Culmination of the Systemic Approach of the Natural Environment"

_systems, doi:10.3390/systems9010008_

Round 1

Reviewer 1 Report

The paper must be revised according with the attached suggestions

Author Response

"I suggest that all of this go as an appendix.On the other hand, the text should be more descriptive."

I put the calculus relative to paragraph "Application of  Newton's law" in appendix C and I tried to make the text more descriptive and comprehensive concerning the new concepts in thermodynamics that I used, notabily the variables and their units. 

"Lastly, there must be conclusions. They do not exist. And it could not be conceived that a methodological issue of importance like this is addressed, without some conclusions appearing."

I totally agree with you and I followed your recommendation

Reviewer 2 Report

Dear Authors

Please see the attached file for my comments

Author Response

That is always the case with a new theory, but the authors make it extremely difficult by not defining the variables and not providing the units of the
variables.

I added a new figure (figure 2) and a new table of the variables used currently. As for the new thermodynamic variables, I improved the figure (now fig.3) and remade the Table 2 presenting the new variables and their units.

" The theory in the article is tested with a small homogeneous soil sample in
the laboratory and it has very little to do with the natural environment where the soils are anything but homogenous. There is no proof that this theory performs well in the natural environment.

Natural environment means here natural coditions of temperature, humidity, pressure of the medium, in the soil as well as at its surface, at the scale of the pedostructure. The question of homogeneity of the natural organisations is an essential question that found its solution in the systemic approach of the hydrostructural pedology.

"Comment 1" ... line 78 and more, about Wmi

It is understandable that you could not understand the cited paragraph if you never have heard about "pedostructure" and its two types of water: intra aggegate (micro) and inter-agegate (macro). This notion exists from a while but stays not very known. In 2014 we found the exact physical equation of the characteristic soil water retention curve. This equation could not have been found without the consideration of these two types of water that are in equilibrium of pressures. It is why we asked for a change of paradigm in soil science already in 2009 (See our article in Global Planetary Change Journal).

"Comment 2 and 3"

That concerns the new vision of the thermodynamics that was published just few weeks before this paper. I took account of your remarks to improve the presentation of the important results of this previous paper, that we absolutly need for conclude our paper. This is recalled and explained in the discussion section and the conclusion (new)  

Round 2

Reviewer 1 Report

Dear authors. The text of this version has improved a lot, and therefore I congratulate you. I especially like the current layout of the intro very much. Figure 1 is key. The paragraph between lines 56 and 64 should be better explained, marking what are the two basic objectives of this paper. Sections two and three have improved a lot. In my opinion, formulas have been downloaded that are better as an appendix. It is much better understood, and makes reading more understandable. The discussion is correct. Just add that I would like the conclusions (L704-L714) to be better structured. Have the objectives been achieved? What is the contribution of this paper to soil hydrodynamics? Is it exportable to all types of soils? With a minimum effort of writing and structuring they can be improved because the study deserves it In my opinion, the conclusions are poor and detract from a very good study.

I recomended to accept it after a little modification.
Yours

Author Response

Thank you very much for your valuable review. Following your advice we clarified the paragraph between lines 56 and 64 and changed the conclusion.

File in attachement

Reviewer 2 Report

Dear Authors

My critical review of your manuscript is attached.  I will not reject the manuscript for publication because I do not understand the theory. However, the manuscript should follow the publication standards and it should be understandable to the reader. This is the reason for the recommendation of "reconsider after major revisions

Regards

Author Response

Thank you for your review

My response in attached file
